# Nanoporous Titanium Enriched with Calcium and Phosphorus Promotes Human Oral Osteoblast Bioactivity

**DOI:** 10.3390/ijerph19106212

**Published:** 2022-05-20

**Authors:** Tania Vanessa Pierfelice, Emira D’Amico, Giovanna Iezzi, Adriano Piattelli, Natalia Di Pietro, Camillo D’Arcangelo, Luca Comuzzi, Morena Petrini

**Affiliations:** 1Department of Medical, Oral and Biotechnological Sciences, University G. d’Annunzio of Chieti-Pescara, Via dei Vestini 31, 66013 Chieti, Italy; tania.pierfelice@unich.it (T.V.P.); emira.damico@unich.it (E.D.); gio.iezzi@unich.it (G.I.); natalia.dipietro@unich.it (N.D.P.); camillo.darcangelo@unich.it (C.D.); morena.petrini@unich.it (M.P.); 2School of Dentistry, Saint Camillus International University of Health and Medical Sciences, Via di Sant’Alessandro 8, 00131 Rome, Italy; 3Dental School, University of Belgrade, 11000 Belgrade, Serbia; 4Research Foundation Villa Serena, 65013 Città Sant’Angelo, Italy; 5Villa Serena Private Clinic of Dott. L. Petruzzi, 65013 Città Sant’Angelo, Italy; 6Center for Advanced Studies and Technology (CAST), University G. d’Annunzio of Chieti-Pescara, 66013 Chieti, Italy; 7Independent Researcher, 31020 Conegliano, Italy; luca.comuzzi@gmail.com

**Keywords:** microroughness, calcium treatment, osteoblasts, nanotopography, dental implants, titanium surface, surface modifications

## Abstract

Implant surfaces are known to influence the osseointegration process; therefore, their modifications represent an important subject of investigation. On this basis, the purpose of this study was to evaluate the response of human oral osteoblasts (hOBs) to three different GR4 titanium discs: Machined, double-etched (Osteopore), and double-etched, surface-enriched with calcium and phosphorus (CaP) (Nanopore). The superficial topography was investigated with scanning electron microscopy (SEM) and the sessile drop technique. To test cellular response and osteoinductive properties, the following points were evaluated: (i) proliferation by MTS assay after 2 and 5 days; (ii) adhesion by multiphoton microscopy at day 2; (iii) the interaction with Ti discs by blue toluidine staining at day 5; (iv) alkaline phosphatase (ALP) activity by ALP assay after 14 days; (v) calcium deposition by alizarin red staining and by cetylpyridinium chloride after 14 days. The SEM analysis showed that Nanopore and Osteopore surfaces were characterized by the same micro-topography. Nanopore and Osteopore discs, compared to Machined, stimulated higher osteoblast proliferation and showed more osteoinductive properties by promoting the ALP activity and calcium deposition. In conclusion, the CaP treatment on DAE surfaces seemed to favor the oral osteoblast response, encouraging their use for in vivo applications.

## 1. Introduction

In recent years, scientific efforts have been mainly focused on changes in surface topography at micro and nano levels in order to alter growth, metabolism, and migration as well as cytokine and growth factor production of osteogenic cells [1]. Although dental implants are available in different materials, shapes, diameters, and lengths, surface features of titanium fixtures play decisive roles for molecular interactions and cellular response of both mammalian and bacterial cells [2,3]. Our recent study demonstrated that the macro, micro, and nano topographies of dental implant surfaces are all crucial for adhesion of fibroblasts during the initial phase of implantation [4]. Many dental implants on the market have a microroughness ranging from 1 to 100 μm, created by manufacturing techniques such as machining, grit-blasting, sandblasting, acid-etching, anodization, and different coating procedures, in order to promote and accelerate the osseointegration [5]. Micro scale modifications contribute to an increase in surface area, which leads to a favorable bone response with increased bone-to-implant contact (BIC) [6]. Other studies showed that surfaces with nanoscale features display additional biological in vitro and in vivo effects [7,8,9,10,11,12]. Ma L et al. [7] suggested that implants with nanofeatures stimulate human mesenchymal stem-cell-derived vesicles to induce osteogenic differentiation. Dabare PRL et al. [8] recently demonstrated that nano topography mediates cell migration by affecting the recruitment of receptor and adapter proteins responsible for cell–surface interaction. In addition, this study showed that nano topography attenuates the proinflammatory tumor necrosis factor alpha (TNF-α) expression. Lopes HB et al. [9] showed that nano topography of dental implants influences cell–implant interactions at the protein level through regulation of specific integrins. Indeed, changes in nano topography occur at physical and chemical levels to exert their effects at the biological level, resulting in influenced osteoblast activity and thereby potentially promoting osseointegration [13]. Osteoblasts are the key cells of the osseointegration process, promoting the deposition of bone matrix [14]. Moreover, the surface energy is mainly affected by the surface chemistry, and only a lower influence is shown by surface roughness alterations [15]. The variety of microstructural and chemical properties that are possible for titanium surfaces opens up opportunities for modifying dental implants to enhance their biological yield.

Indeed, the purpose of this study was aimed to compare the response of human oral osteoblasts on three different titanium surfaces: (i) Machined; (ii) double-etched (Osteopore); (iii) double-etched enriched with calcium and phosphorus (CaP) treatment (Nanopore), through the evaluation of cell proliferation, adhesion, interaction with Ti discs, ALP activity and calcium deposition.

## 2. Materials and Methods

### 2.1. Dental Implant Discs

Titanium discs were provided by AoN Implants (Grisignano di Zocco, Italy), corresponding to the commercially pure titanium (cpTi) GR4. All tested discs were sized 5 mm in diameter and 2 mm in thickness to fit securely in a 96-well plate. The sample discs were divided into 3 groups:
-Machined implant surface with no roughening or CaP treatments applied in cold-drawn titanium GR4;-Osteopore implant surface was treated by double acid-etching in order to create surface structures and roughness at the micro level. This treatment was followed by washing and final decontamination by plasma;-Nanopore implant surface was treated by double acid-etching in order to create surface structures and roughness at the micro level and were further enriched in calcium and phosphorus (CaP treatment) by process of inorganic salts in aqueous solution, to superimpose a nanotopographic complexity. This treatment was followed by washing and final decontamination by plasma.

Machined surfaces were considered as control.

Prior to performing experiments, all samples were sterilized. Primarily, to remove synthetic oils used for processing, the samples were subjected to a degreasing process with diluent. Subsequently, dental implant surfaces were washed by ultrasound in a solution of demineralized water and soap, rinsed, and washed again with distilled water. At the end of the acid treatments and the CaP enrichment, a final decontamination is carried out using argon plasma. Concluding sterilization was performed with gamma-ray exposure.

This study follows the appropriate EQUATOR guidelines and the Standards for Reporting Qualitative Research SRQR.

### 2.2. Scanning Electron Microscopy (SEM) Analysis

Surface topography was investigated by Phenom Desktop SEM (Phenom-World BV, Eindhoven, The Netherlands). Five discs of each group were mounted on aluminum stubs with conductive glue. Images were taken using an accelerating voltage of 15 kV with the backscattered electronic signal detector (BSE), BSD full, at 290× and 1200× magnification.

### 2.3. Water Contact Angle Measurement

The surface wettability was investigated using the sessile drop technique, as previously described [3,16]. Briefly, 1 µL of saline solution was pipetted on each disc and immediately a Nikon D90 DSLR camera (Nikon Corporation, Tokyo, Japan) with an 18–105 mm lens was used to photograph the samples. The water contact angle was then measured using ImageJ 1.52 q for Mac OS X (National Institute of Health, Bethesda, MD, USA).

### 2.4. Oral Osteoblasts Culture

Primary oral osteoblasts (hOBs) were isolated from mandible bone fragments of n° 12 patients that underwent the surgical removal of lower third molars at the dental clinic of the G. D’Annunzio University. All patients signed an informed consent in accordance with the Declaration of Helsinki principles and according to the ethical standards of the Institutional Committee on Human Experimentation (reference number: BONEISTO N. 22 10 July 2021). Immediately after sampling, each bone fragment underwent three enzymatic digestions at 37 °C for 20, 30 and 60 min utilizing a solution consisting of collagenase type 1A (Sigma-Aldrich, St. Louis, MO, USA) and trypsin-EDTA 0.25% (Sigma-Aldrich) dissolved in Dulbecco’s Modified Eagle’s medium (DMEM, Corning, New York, NY, USA) at 10% fetal bovine serum (FBS, Gibco-Life Technologies, Monza, Italy). The solution obtained from the enzymatic digestion was centrifuged at 1200 rpm for 10 min. Then, the pellet obtained was transferred into a T25 culture flask with low-glucose (1 g/L) DMEM supplemented with 10% FBS, 1% antibiotics (100 µg/mL^−1^ streptomycin and 100 IU/mL^−1^ penicillin), and 1% L-glutamine to promote a final spontaneous migration of the cells. The isolated hOBs were cultured at 5% CO_2_ and 37 °C to achieve their confluence to be used between the 3rd and the 5th passage upon the characterization by cytometric analysis. Following 10 days of culture, bone fragments were removed.

### 2.5. Proliferation Study

Primary human osteoblasts were cultured on titanium discs and their proliferation was assessed by CellTiter96-assay (3-(4,5-dimethylthiazolyl-2)-2,5-diphenyltetrazolium bromide) after 2 and 5 days, according to the manufacturer’s instruction (MTS, Promega, Madison, WI, USA). Briefly, 1 × 10^4^ cells/disc were seeded onto the surfaces of the titanium discs put in 96-wells/plate and were incubated for 2 and 5 days at 37 °C and 5% CO_2_. After incubation time, 10 µL of MTS solution was added to each culture well for 2h of incubation. The absorbance, as optical density (OD) values, was detected at 490 nm using a spectrophotometer microplate (Synergy H1 Hybrid BioTek Instruments, Winooski, VT, USA). The number of cells was extrapolated starting from the OD values.

### 2.6. Multiphoton Microscopy

The hOBs adhesion on titanium surfaces was evaluated by multiphoton microscopy. 1 × 10^4^ cells/disc were seeded on the top of discs. After 2 days of culture at 37 °C and 5% CO_2_, adherent cells were rinsed twice with PBS and fixed with 4% paraformaldehyde for 10 min. Osteoblasts were washed twice with cold PBS, permeabilized with 0.1% Triton 100× in PBS for 10 min and incubated with DAPI solution (1:1000 in PBS). The discs were observed by multiphoton microscopy LSM 710 (Zeiss, Oberkochen, Germany) with a filter SP < 485 nm at 500–550 nm. Images were taken at 7× and 40× magnification.

### 2.7. Cell Staining

To evaluate the interaction between the titanium surfaces and hOBs, toluidine blue staining was performed. A total of 2 × 10^4^ cells/well were cultured on the bottom of the plate well with the disc for 5 days at 37 °C and 5% CO_2_. The discs were then removed, and adherent cells were stained with 1% toluidine blue and 1% borax, upon two washings with PBS and fixation with 70% cold ethanol. Cells were then observed by a stereomicroscopy connected with a camera at a magnification of 6× and 25× (Leica, Wild Heerbrugg, Wetzlar, Germany).

### 2.8. ALP Assay

Alkaline phosphatase (ALP) activity was determined to evaluate the function of osteoblasts after their culture on titanium discs. ALP was evaluated by ALP assay kit colorimetric AB83369 (Abcam Inc., Cambridge, UK) that is based on the cleavage of p-nitrophenyl phosphate (pNPP). Briefly, 5 × 10^4^ cells/disc were seeded on Ti surfaces in 24-well culture plates for 14 days. Every 3 days, fresh medium was added. After 14 days of culture, osteoblasts seeded on Ti discs were collected, and cell lysate was obtained by homogenizing cell suspension through Tissue Rupture device (QIAGEN, Hilden, Germany), previously washed three times with PBS and resuspended in assay buffer. After centrifuging at 10,000× *g* for 15 min, the supernatant was collected, and the relative ALP activity was measured in agreement with manufacturer instructions. The absorbance values were measured at 405 nm.

### 2.9. Alizarin Red Staining

The influence of surface features of titanium discs on the mineralization capability of human osteoblasts was qualitative evaluated by alizarin red staining (ARS). In brief, cells were seeded at a density of 5 × 10^4^ cells/disc onto the surface of discs into the 24-well culture plate. After 14 days of culture, titanium discs were removed, and adhered cells on the bottom of the plates were rinsed three times with PBS and fixed with glutaraldehyde solution (2.5%) for 2 h. A total of 1 mL of AR staining solution (Sigma-Aldrich) was then added for 1 h at room temperature, and deionized water was used to remove the excess dye. The presence of calcium deposition was qualitatively evaluated by observing the intensity of the red color. Images were taken by a stereomicroscope connected with a camera at a magnification of 12× (Leica, Wild Heerbrugg, Wetzlar, Germany).

### 2.10. Calcium Deposition

To quantify calcium deposition, cetylpyridinium chloride (CPC) was added to the bottom of the plate after the qualitative observation upon ARS, as reported in the previous paragraph. Well plates were washed with deionized water and were treated with 1 mL of 10% CPC solution (Sigma-Aldrich) for 1 h to chelate calcium ions. After incubation, the absorbance was read at 540 nm in a microplate reader (Synergy H1 Hybrid BioTek Instruments) and normalized with cell number.

### 2.11. Statistical Analysis

All experiments were performed in biologic triplicates and repeated three times. The data are reported as means ± standard deviation (SD). Statistical analyses were performed using the GraphPad Prism8 (GraphPad Software San Diego, CA, USA). The Levene Test permitted the confirmation of the homogeneity of the groups tested; then, the ANOVA and post hoc Tukey tests were adopted. A *p*-value < 0.05 was considered as significant.

## 3. Results

### 3.1. Surface Characterization

The SEM observation at 290× (Figure 1) showed the microscopical features of the samples: Machined surfaces (Figure 1A) were characterized by a very regular structure with the presence of circumferential and parallel lines; Nanopore and Osteopore surfaces showed the same topography, characterized by the loss of turning lines and the presence of an increased roughness with a random disposition. Viewed at 1200×, the microstructures created by etching treatment were visible on the Nanopore and Osteopore surfaces (Figure 1E,F) that are lacking on the Machined surfaces (Figure 1D). In particular, Osteopore and Nanopore showed micro-irregularities such as pits. The presence of calcium and phosphorus (CaP) covering the titanium Nanopore substrate leaves the underlying original microtopography of the surface clearly visible in Figure 1E.

From the sessile drop technique, all tested surfaces resulted hydrophilic with contact angles less than 90° (Figure 2A). The sessile drop method permitted the measurement of the water contact angle (WCA) of the different surfaces (Figure 2B): Machined (34.53°), Nanopore (53.86°), and Osteopore (63.60°). The ANOVA test resulted *p* < 0.0001. Nanopore and Osteopore surfaces showed significantly larger contact angles than Machined (Machined vs. Nanopore *p* = 0.0001; Machined vs. Osteopore *p* < 0.0001). Thus, the wettability was altered by double-etching and CaP treatments.

### 3.2. Osteoblasts Proliferation

Results of proliferation measurement by the MTS assay on the second and fifth day are shown in Figure 3. At two time points, all tested surfaces showed the number of cells greater than the number of seeded cells. On the second day, cell numbers were similarly upregulated on Nanopore and Osteopore surfaces, +17.49 ± 0.02% and +17.83 ± 0.03% respectively, compared to the non-treated Machined surface, but results did not reach statistical significance. On the fifth day, the CaP-treated Nanopore surface showed a significant increase in cell population with +36.53 ± 0.13% relative to the non-treated smooth Machined surface and showed a greater number of cells +20.79 ± 0.13% than the non-treated microrough Osteopore surface (Machined vs. Nanopore *p* = 0.0001; Nanopore vs. Osteopore *p* = 0.001). The ANOVA test revealed *p* = 0.0008.

### 3.3. Adhesion of Osteoblast on Titanium Discs

Human osteoblasts showed adhesion on all surfaces tested in this study. Cells after 2 days of culture appeared to attach and spread more on the microrough Nanopore and Osteopore surfaces in comparison to smooth Machined surface (Figure 4A–C). Between the two microrough surfaces, Osteopore discs presented a higher number of nuclei stained by DAPI with respect to Nanopore discs. No multinucleated cells were detected under multiphoton microscopy assessment of cells attached to all surfaces (Figure 4D–F).

### 3.4. Interaction between Osteoblasts and the Tested Surfaces

The interaction between osteoblasts and titanium surfaces was evaluated by staining cells seeded on the bottom of plate around discs. Cells mainly interacted with Nanopore discs (Figure 5B,E) with respect to Machined (Figure 5A,D) and Osteopore (Figure 5C,E). The Nanopore disc seems to attract a higher number of osteoblasts. At higher magnification, a typical shape and morphology of osteoblasts can be observed (Figure 5D–F).

### 3.5. Alkaline Phosphatase Activity

At day 14, ALP activity enhanced significantly by +64.97 ± 0.05% in cells cultivated on the top of Nanopore surfaces compared to Machined discs (*p* < 0.0001). Similarly, ALP was significantly stimulated when cells were seeded on Osteopore surfaces, despite a lower level being observed with respect to Nanopore (Machined vs. Nanopore *p* < 0.0001; Machined vs. Osteopore *p* < 0.0001; Nanopore vs. Osteopore *p* < 0.0001) (Figure 6). The ANOVA test resulted in *p* < 0.0001.

### 3.6. Calcium Deposition

The matrix mineralization was qualitatively evaluated at 14 days of culture by alizarin red staining. Mineralization was stimulated by Nanopore surfaces, as shown by calcified nodules that appear denser and brighter red. In comparison, the calcium deposition observed for Osteopore resulted as more similar to Machined (Figure 7A). The observation from AR staining was confirmed by the quantitative results, obtained through the measurement with CPC. The histogram showed the highest calcium deposition percentages for Nanopore, followed by Osteopore (Figure 7B).

## 4. Discussion

Long-term success of titanium implants depends on their effective osseointegration [17], which is directly related to the interaction of osteoblasts with titanium surfaces [18]. Therefore, the evaluation of the behavior of osteoblasts on titanium surfaces, as well as the molecular events, are crucial to determine the influence of different surface characteristics on oral implantology.

In this study, human oral osteoblasts were cultured on three titanium surfaces: Machined, Nanopore and Osteopore. In particular, Nanopore and Osteopore surfaces have the same micro-topographical patterns induced by the double-etching process and enrichment. In addition, Nanopore underwent an enrichment with CaP. Machined, which lacks the double-etching process with CaP enrichment, was used as a control. The effects of these surface characteristics on cellular proliferation, material interaction, adhesion, and osteogenic induction were investigated.

The previous literature showed an improved implant success rate in correlation with the increase in surface roughness and wettability [13,19]. In our study, the effects of DAE treatment on titanium Nanopore and Osteopore discs determined an increased micro-roughness that resulted in a similar topography at SEM observation. These surfaces were both characterized by higher roughness than Machined surfaces and by randomly disposed pits. However, surprisingly, these surfaces were less hydrophilic than Machined surfaces. A previous investigation has demonstrated that microroughness gives an initial hydrophobic configuration probably due to air entrapped in the smallest micropores, which can shift to a hydrophilic one with time [20]. The surface modification of Nanopore by the CaP treatment resulted in a higher hydrophilicity of Nanopore with respect to Osteopore. Machined surfaces resulted in being the most hydrophilic. The previous literature showed that bone growth is encouraged at the cellular level by microroughness that attracts differentiating osteogenic cells [13,19]. The effects of DAE treatment on titanium Nanopore and Osteopore discs that determined an increased micro-roughness resulted in a more favorable impact on the proliferative activity of osteoblast cells at 2 and 5 days rather than a smooth surface. It is worth noting that the early biological response was very similar between the micro roughed Osteopore and Nanopore surfaces. It is important to highlight that the synergic effect of CaP and DAE treatments in Nanopore significantly promoted cell proliferation at 5 days, more so than Machined and Osteopore. This is in line with a recent in vivo study in which the effect of nano topography superimposed on microtopography was confined to the early period of woven bone formation and the concomitant remodeling (6–21 days) [8].

The oral osteoblast adhesion was analyzed 48 h after seeding because this process on planar surfaces usually occurs from 36 to 48 h [21]. The acid-etching step applied to create the micro-irregularities on the surface resulted again in increased functional surface area and improved bio adhesion of Osteopore and Nanopore discs with respect to the smooth Machined surface. Between micro-roughed surfaces, Osteopore showed a slightly higher density of adhered osteoblasts than Nanopore. This may suggest that surface topography and wettability differently influence the attachment and the proliferation rate of cells. The study of the interaction between cells and dental implant discs showed that the chemical-driven effect encouraged cell migration towards Nanopore implants, revealing a greater affinity of osteoblast for surfaces that have CaP enrichment. This is in accordance with a recent study that demonstrates how the gap closure between cell monolayers is faster on surfaces having modifications at the nanoscale level and how this phenomenon is predominantly driven by cell migration [8]. As is known, chemical modifications alter the surface energy in addition to the surface chemistry. These changes in surface features influence protein adsorption, which in turn changes the cell response to a material [19]. An animal study in 2009 showed how the addition of calcium phosphate particles on titanium implants has better osteoconduction properties compared to commercially pure titanium (cpTi) and displayed increased bone-to-implant contact due to an enhancement in surface nanotopography [22]. Another study demonstrated that the surfaces enriched with calcium phosphate particles showed a significant reduction in bacterial attachment compared to Machined and acid-etched implant surfaces [23]. In our study, osteoblasts seeded onto a CaP-enriched Nanopore disc also showed the most enhancement of ALP activity and mineral nodule deposition, which are in vitro indicators of the mineralization capacity of these cells. These abilities, assessed at 14 days, resulted in increased cells seeded on the Osteopore surface when compared to the Machined surface. Altogether, the results of this study suggest that physical and chemical modifications could show significant results in a very short time after implant placement; however, modifications might require between two and five days to obtain the osteoblast response. This suggests that the time frames are important to analyze and fully understand the benefits of surface modifications. The results also suggest that Ti implants with combined nanoscale and microscale surface features have more favorable bone responses than Machined. Moreover, the synergistic work between the nanotopography and the underlying microroughness results in a range of beneficial effects, and it is essential to consider at what point in time and under what conditions these effects occur.

## 5. Conclusions

The Nanopore surfaces, characterized by double-etching treatment followed by CaP enrichment, resulted in increased proliferation, adhesion, alkaline phosphatase activity and calcium deposition compared to Machined and Osteopore discs. This study may represent a helpful step for clinicians in the in vivo use of these novel surface-characterized osteoconductive and osteoinductive properties.

## Figures and Tables

**Figure 1 ijerph-19-06212-f001:**
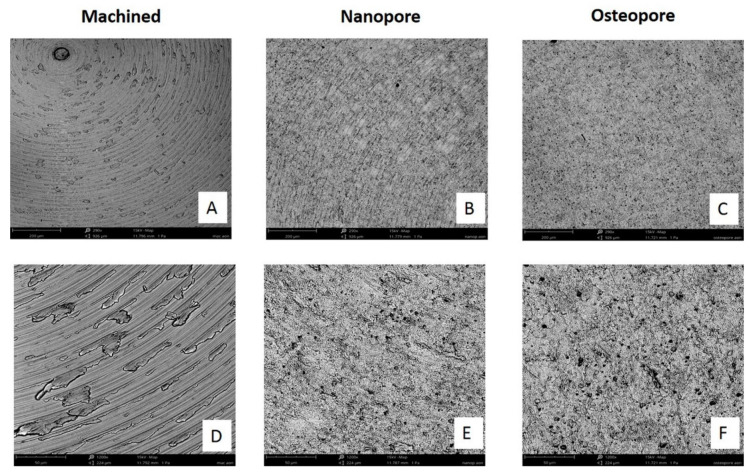
SEM observations of Ti discs. Topographic macro characteristics were evidenced at low magnification of 290× (**A**–**C**). Micro-topographic features of the discs were revealed at the magnification of 1200× (**D**–**F**). The CaP enrichment of nanopore surface did not change the topography at the micro level, because the original underlying metal microtopography was still preserved and could be easily identified (**E**). (Scale bar: 200 μm at low magnification, 50 μm at high magnification).

**Figure 2 ijerph-19-06212-f002:**
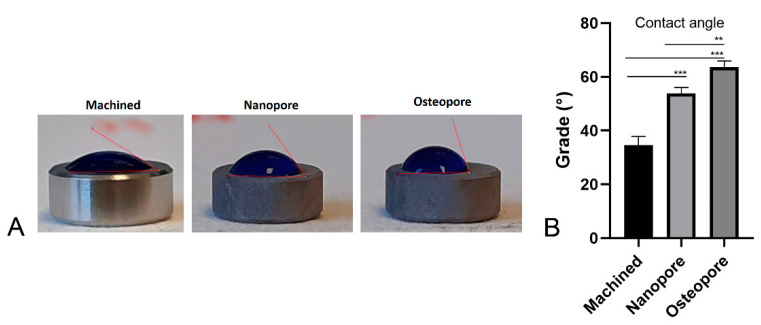
Water contact angle measurement. (**A**) Pictures of the WCA on the different surfaces, during the sessile drop method. (**B**) Water contact angle values (** *p* < 0.001; *** *p* ≤ 0.0001; Machined vs. Nanopore *p* = 0.0001; Machined vs. Osteopore *p* < 0.0001; Nanopore vs. Osteopore *p* < 0.001).

**Figure 3 ijerph-19-06212-f003:**
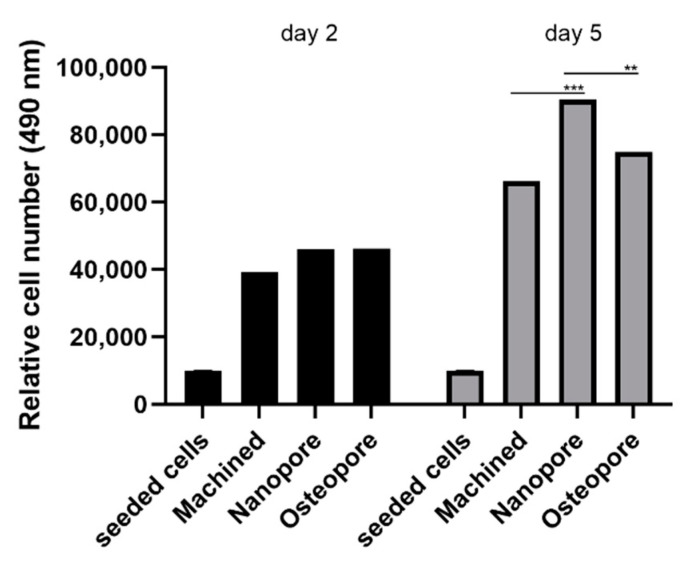
Cell proliferation analysis of osteoblasts seeded on Ti discs. The histogram shows the enhanced proliferation for cells cultured on Nanopore (+36.53 ± 0.13%) and Osteopore (+13.03 ± 0.19%) surfaces with respect to Machined at day 5. Data are expressed in the form of relative cell number seeded on the discs as mean ± SD. All standard deviations resulted between 0.02 and 0.19. (** *p* ≤ 0.001; *** *p* ≤ 0.0001; Machined vs. Nanopore *p* = 0.0001; Nanopore vs. Osteopore *p* = 0.001).

**Figure 4 ijerph-19-06212-f004:**
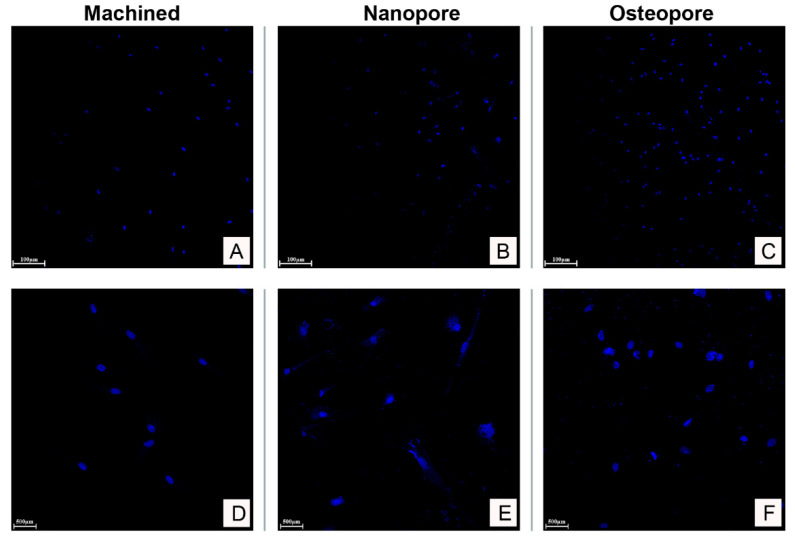
Primary osteoblasts cultured on tested surfaces for 2 days. Cells were fixed in 4.0% paraformaldehyde and nuclei were stained with DAPI (blue). Images were taken by multiphoton microscopy at the magnification of 7× (**A**–**C**) (Scale bar: 100 μm) and 40× (**D**–**F**) (Scale bar: 500 μm).

**Figure 5 ijerph-19-06212-f005:**
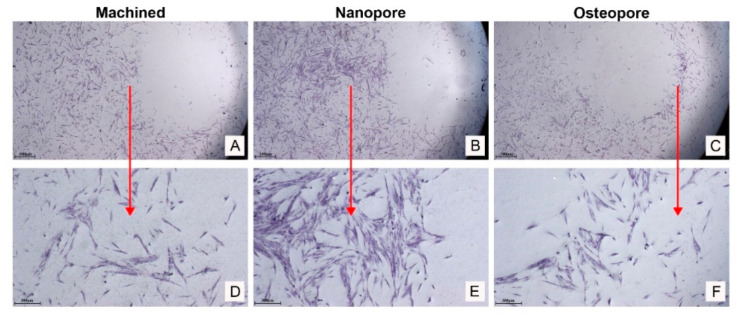
Osteoblasts differently interacted with Ti discs. The enrichment in calcium and phosphorus promoted the interaction between primary osteoblasts and Nanopore surface at late stage of proliferation (5 days). Images were taken by an inverted microscope connected with a camera at the magnification of 6× (**A**–**C**) and 25× (**D**–**F**). The red arrows indicate the magnification of cell images (Scale bar: 100 μm at low magnification and 300 μm at higher magnification.)

**Figure 6 ijerph-19-06212-f006:**
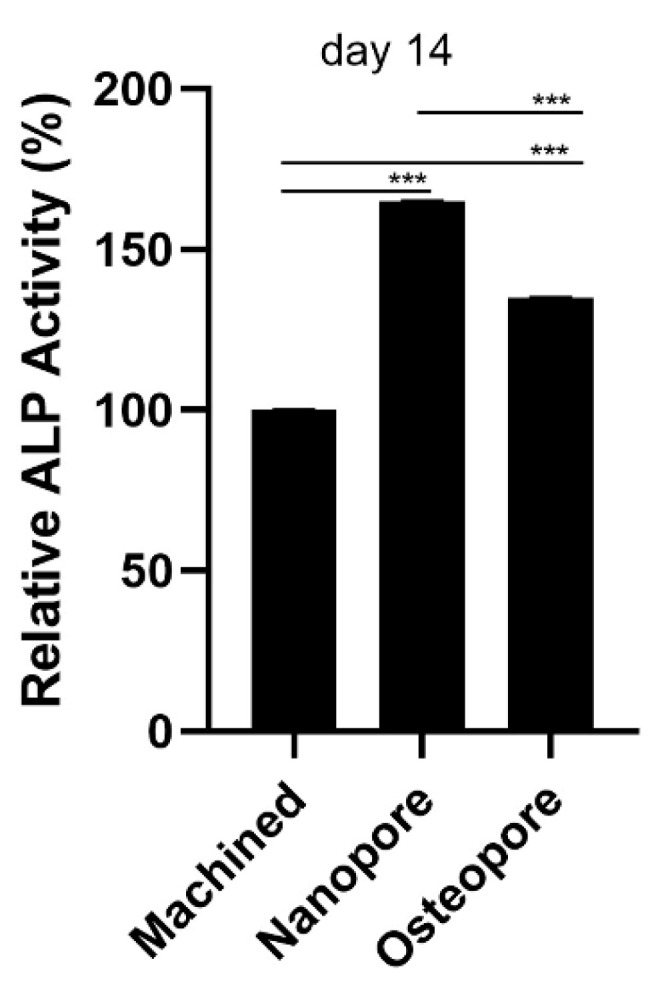
ALP activity of primary human osteoblasts cultured after 14 days. Nanopore and Osteopore surfaces showed higher ALP activity than Machined, +64.97 ± 0.05% and +34,92 ± 0.09% respectively. Data are expressed with relation to cell percentage cultivated on Machined disc as mean ± SD. Standard deviation values resulted between 0.01 and 0.09. (*** *p* < 0.0001; Machined vs. Nanopore *p* < 0.0001; Machined vs. Osteopore *p* < 0.0001; Nanopore vs. Osteopore *p* < 0.0001).

**Figure 7 ijerph-19-06212-f007:**
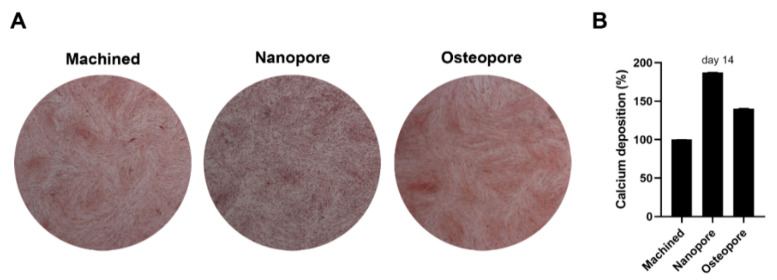
Effects of surface modifications on mineralization. (**A**) CaP enrichment sub-monolayers of Nanopore discs enhanced calcium deposition evaluated by alizarin red staining. Images were taken at magnification of 12× by an inverted microscopy connected with a camera. (**B**) Quantitative measurement was performed with cetylpyridinium chloride. Data are expressed in relation to cell percentage cultivated on machined disc.

## Data Availability

Data supporting reported results can be found, including links to publicly archived datasets analyzed or generated during the study.

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
