# Peer review of "Nanoporous Titanium Enriched with Calcium and Phosphorus Promotes Human Oral Osteoblast Bioactivity"

_ijerph, 2022, doi:10.3390/ijerph19106212_

Round 1

Reviewer 1 Report

The manuscript, entitled

Nanoporous titanium enriched with calcium and phosphorus enhance human oral osteoblasts bioactivity

The authors tried to evaluate the response of human oral osteoblasts to three different GR4 titanium discs: machined, osteopore, and nanopore.

I have a few suggestions related to the manuscript- 

Other studies showed that surfaces with nanoscale features display additional biological in vitro and in vivo effects. No reference has been added after this sentence. I would like to request that the authors add at least three references, which include vitro and vivo studies. 

Was any coating applied to the five discs of each group before the SEM image was taken? If so, the author should mention it.

All the sections are clearly described, and the manuscript has a significant impact on the dental society. The manuscript can be accepted.

Author Response

The manuscript, entitled

Nanoporous titanium enriched with calcium and phosphorus enhance human oral osteoblasts bioactivity

The authors tried to evaluate the response of human oral osteoblasts to three different GR4 titanium discs: machined, osteopore, and nanopore.

I have a few suggestions related to the manuscript- 

Other studies showed that surfaces with nanoscale features display additional biological in vitro and in vivo effects. No reference has been added after this sentence. I would like to request that the authors add at least three references, which include vitro and vivo studies. 

Thank you for your suggestion. We added the references at the end of this sentence:

  1. Ma, L.; Li, G.; Lei, J.; Song, Y.; Feng, X.; Tan, L.; Luo, R.; Liao, Z.; Shi, Y.; Zhang, W.; et al. Nanotopography Sequentially Mediates Human Mesenchymal Stem Cell-Derived Small Extracellular Vesicles for Enhancing Osteogenesis. ACS nano 2021, 16, 415–430, doi:10.1021/ACSNANO.1C07150.
  2. Dabare, P.R.L.; Bachhuka, A.; Visalakshan, R.M.; Shirazi, H.S.; Ostriko, K.; Smith, L.E.; Vasilev, K. Mechanistic Insight in Surface Nanotopography Driven Cellular Migration. ACS biomaterials science & engineering 2021, 7, doi:10.1021/ACSBIOMATERIALS.1C00853.
  3. Lopes, H.B.; Freitas, G.P.; Fantacini, D.M.C.; Picanço-Castro, V.; Covas, D.T.; Rosa, A.L.; Beloti, M.M. Titanium with Nanotopography Induces Osteoblast Differentiation through Regulation of Integrin ΑV. Journal of cellular biochemistry 2019, 120, 16723–16732, doi:10.1002/JCB.28930.

Was any coating applied to the five discs of each group before the SEM image was taken? If so, the author should mention it.

To evaluate the microtopography of discs at SEM we did not use any coating. We usually coat discs with gold when we observe cells cultured on discs.

All the sections are clearly described, and the manuscript has a significant impact on the dental society. The manuscript can be accepted.

Reviewer 2 Report

Thank You for sending article titled “Nanoporous titanium enriched with calcium and phosphorus enhance human oral osteoblasts bioactivity” from Int. J. Environ. Res. Public Health (MDPI) for review.

Comments:

Interesting article and thank you for submitting your work. 

MAJOR POINT: In addition to the patient's signature, the consent of the Bioethics Committee is required to collect a bone fragment from the mandible for examination.

Author Response

Thank You for sending article titled “Nanoporous titanium enriched with calcium and phosphorus enhance human oral osteoblasts bioactivity” from Int. J. Environ. Res. Public Health (MDPI) for review.

Comments:

Interesting article and thank you for submitting your work. 

MAJOR POINT: In addition to the patient's signature, the consent of the Bioethics Committee is required to collect a bone fragment from the mandible for examination.

Thank you for your observation, the ethical committee approval (BONEISTO N. 22 del 10/07/2021) is indicated in Oral Osteoblasts Culture in Material and Methods.

Reviewer 3 Report

Dear Authors,

The article: 'Nanoporous titanium enriched with calcium and phosphorus enhance human oral osteoblasts bioactivity' was to evaluate the response of human oral osteoblasts (hOBs) to three different GR4 titanium discs: Machined, double etched (Osteopore), and double etched surface enriched with Calcium and Phosphorus (CaP) (Nanopore). 

English language and style are fine.

Punctuation mistakes should be corrected. 

Order the affiliation numbering.

line 53: 'Ma L et al. suggested that implants with nanofeatures stimulate human mesenchymal stem cell-derived vesicles to induce osteogenic differentiation [7].' should be 'Ma L et al. [7] suggested that implants with nanofeatures stimulate human mesenchymal stem cell-derived vesicles to induce osteogenic differentiation.' Apply this style throughout your work.

Materials and methods

The materials and the method are well described and very reliable.

The p value should be in italics.

Results

line 252 skip to another page

The p value should be in italics.

Discussion is clearly presented.

Add table with abbeviations befeore references.

References should be prepared in accordance with the MDPI guidelines.

To sum up, article can be accepted after minor revison. 

Author Response

Dear Authors,

The article: 'Nanoporous titanium enriched with calcium and phosphorus enhance human oral osteoblasts bioactivity' was to evaluate the response of human oral osteoblasts (hOBs) to three different GR4 titanium discs: Machined, double etched (Osteopore), and double etched surface enriched with Calcium and Phosphorus (CaP) (Nanopore). 

English language and style are fine.

AUTHOR’S ANSWER: Thank you.

Punctuation mistakes should be corrected. 

AUTHOR’S ANSWER: Thank you for your observation. We reviewed the punctation.

Order the affiliation numbering.

AUTHOR’S ANSWER: Thank you for your observation. We corrected the affiliation. Unfortunately, this mistake was due to the name of an author “Luca Comuzzi” that was missing. Thus, we provided to introduce him in this reviewed version of the manuscript.

line 53: 'Ma L et al. suggested that implants with nanofeatures stimulate human mesenchymal stem cell-derived vesicles to induce osteogenic differentiation [7].' should be 'Ma L et al. [7] suggested that implants with nanofeatures stimulate human mesenchymal stem cell-derived vesicles to induce osteogenic differentiation.' Apply this style throughout your work.

AUTHOR’S ANSWER: Thank you for your suggestion. We adopted this style.

Materials and methods

The materials and the method are well described and very reliable.

The p value should be in italics.

AUTHOR’S ANSWER: Thank you for your observation. We used the italics for p value

Results

line 252 skip to another page

The p value should be in italics.

AUTHOR’S ANSWER: Thank you for your suggestions

Discussion is clearly presented.

Add table with abbeviations befeore references.

AUTHOR’S ANSWER: Thank you for your suggestion. We have introduced a table with abbreviations

References should be prepared in accordance with the MDPI guidelines.

AUTHOR’S ANSWER: Thank you for your observation. We conformed the references to MDPI guidelines.

To sum up, article can be accepted after minor revision. 

AUTHOR’S ANSWER: Thank you for your observation.

Reviewer 4 Report

GENERAL COMMENTS

The article claims to have tested biocompatibility/cytotoxicity, but the tests were not performed in accordance with ISO 10993 and ISO 7405 standards, this these tests and conclusions were not reliable and must be deleted and the words used to describe these tests must be revised to state “proliferation.” Proliferation is not the same as biocompatibility.

The introduction and discussions are unhelpful and flawed because they give a poor background to the research. They focus on irrelevant issues not directly related to the methods. It should have explained the functions of osteoblasts, proliferation, biocompatibility, cytotoxicity, cell survival and bioactivity, because these were the tests that were conducted.

There is a lack of controls to help validate the data.

It is not clear if the disks have the same surface properties as commercially available implants.

All the implant disks were supplied by a single manufacturer “AoN” which may indicate some bias towards that supplier.

The methods do not explain how the implant disks were sterilized, but later gives gamma radiation. Did the sterilization heating or gas or abrasion, modify the surface chemistry? Was an aseptic technique used when adding the implants to the cells?

The osteoblast cell lines were grown in DMEM culture with non-heat inactivated 10% fetal calf serum (FCS). Due to the use of FCS containing growth factors the native properties of the cells will be altered and they likely do not have the same bioreactivity of osteoblasts. Thus, it is likely that the cell responses are difference to natural osteoblasts, and this confounds the results.

The images are out of focus, pixilated, and indicate the wrong magnifications, and have text too small to read.

There are too many problems with the quality of this article to recommend its publication. I suggest the authors be allowed to answer the reviewer comments and submit a revised article.

TITLE

  1. Acceptable

ABSTRACT

  1. Page 1, line 30. The incorrect statement must be deleted because no testing conformed to ISO standards for cytotoxicity: “None of the tested surfaces showed any cytotoxicity.”

INTRODUCTION

  1. The introduction needs to provide a background to the tests and results, so that readers can understand the topic. It failed, the introduction must explain the functions of osteoblasts, proliferation, biocompatibility, cytotoxicity, cell survival and bioactivity.
  2. The introduction needs to explain the purpose of this study, similar to the abstract. The purpose described here must be the same as the abstract, it is not and it confuses the reader.

MATERIALS AND METHODS

  1. There is a lack of controls to help validate the data.
  2. It is not clear if the disks have the same surface properties as commercially available implants.
  3. All the implant disks were supplied by a single manufacturer “AoN” which may indicate some bias towards that supplier.
  4. The methods do not explain how the implant disks were sterilized, but later writes gamma ray exposure. Did the sterilization heating or gas or abrasion, modify the surface chemistry?
  5. For the scanning electron microscopy (SEM), why was x1,200 selected? It seems a low magnification for a SEM.
  6. The osteoblast cell lines were grown in DMEM culture with non-heat inactivated 10% fetal calf serum (FCS). Due to the use of FCS containing growth factors the native properties of the cells will be altered and they likely do not have the same bioreactivity of osteoblasts. Thus, it is likely that the cell responses are difference to natural osteoblasts, and this confounds the results. The use of FCS with growth factors must be justified and explained to readers.
  7. The proliferation study is not the same as biocompatibility/cytotoxicity, and there were no controls to help validate the proliferation data.
  8. How does “multiphoton microscopy” differ from scanning electron microscopy (SEM)? This should be described as SEM to avoid confusing readers.
  9. Why were no controls used to validate the cell staining study? I disagree with the removal of the disks, because it likely discarded the attached cells.
  10. It is not clear if the cell culture media was changed during the 14 days of the ALP assay, and no control was used to help validate the accuracy of the data.
  11. I disagree with the removal of the disks from the Alizarin red staining, because it likely discarded the attached cells, and the lack of negative and positive controls to help validate the data.
  12. The statistics were performed with ANOVA, but ANOVA only gives one P value per group. But the graphs show paired group P values, was this T tests? If so, the description of the statistics must be revised.

RESULTS

  1. The results state a 290x SEM magnification which is different to the x1200 SEM magnification in the methods. Which is correct or incorrect?
  2. How did the ANOVA P value give a P value between the three groups Nanopure, Osteopure, and Machined (P<0.001)

DISCUSSION

  1. Unfortunately the discussion is as unfocused, confusing and as unhelpful as the introduction in helping the reader understand the topic and what was done and the significance of the results.
  2. Many of the sentences belong in the introduction as background.
  3. p10, line 338: “Certain modifications” and “other modifications” have unclear meanings.
  4. p10, line 339: The final sentence is confusing and ambiguous to readers “This suggests that further time frames are important to analyze to fully understand the benefits of each variation.”

CONCLUSION

  1. The conclusion is confusing; it should have explained the results more clearly to the reader. This can’t be the first step towards a novel surface, because the surfaces are already novel.

REFERENCES

  1. There are 19 references, most are recent, more references and more text in the introduction would improve the quality of this manuscript.

IMAGES

24.  The images are out of focus, pixilated, and indicate the wrong magnifications, and have text too small to read, as follows:

25. Figure 1A, 1B, 1C and 1F are out of focus, and are not acceptable sharpness for publication

26. Figure 1A. Nanopure is out of focus and not an acceptable sharpness for publication.

27. Figure 2B. Bar chart is pixilated, and the font size is too small in the upper region.

28. Figure 2. Has no error bars, why?

29. Figure 4. I cannot see anything in most of these fluorescent images, and the blue cells are out of focus.

30. Figure 5. The cell culture photographs are out of focus, and not sharp enough for a publication. The x6 magnification is incorrect, it must be about x200, you need to calculate the microscope magnification, the x6 is probably only the lens magnification.

31. Figure 6. Has no error bars, why?

32. Figure 7. The cell culture photographs are out of focus, and not sharp enough for a publication. The x12 magnification is incorrect, it must be about x200, you need to calculate the microscope magnification, the x12 is probably only the lens magnification.

Author Response

GENERAL COMMENTS

The article claims to have tested biocompatibility/cytotoxicity, but the tests were not performed in accordance with ISO 10993 and ISO 7405 standards, this these tests and conclusions were not reliable and must be deleted and the words used to describe these tests must be revised to state “proliferation.” Proliferation is not the same as biocompatibility.

AUTHOR’S ANSWER: Thank you for your observation, but our study is aimed for research purposes. The manufacturer has already assessed the implants in accordance with ISO before introducing them to the market. The objective of our study is to evaluate what surface showed a better cellular response.

The introduction and discussions are unhelpful and flawed because they give a poor background to the research. They focus on irrelevant issues not directly related to the methods. It should have explained the functions of osteoblasts, proliferation, biocompatibility, cytotoxicity, cell survival and bioactivity, because these were the tests that were conducted.

AUTHOR’S ANSWER: Thank you for your observations. We have introduced the following sentences in the introduction:

  • “Osteoblasts are the key cells of the osseointegration process, promoting the deposition of bone matrix [Komori T, 2019]” (lines 67-68).

Furthermore, we have removed all sentences referred to cytotoxicity and biocompatibility assays. In this study we evaluated the proliferation, the adhesion, ALP activity and mineralization.

There is a lack of controls to help validate the data.

AUTHOR’S ANSWER: Thank you for your observation. The control was represented by Machined discs. Machined disc is a surface in titanium GR4 with no roughening or CaP treatments applied. It is used as control in several works:

Velázquez-Cayón R, Castillo-Dalí G, Corcuera-Flores JR, Serrera-Figallo MA, Castillo-Oyagüe R, González-Martín M, Gutierrez-Pérez JL, Torres-Lagares D. Production of bone mineral material and BMP-2 in osteoblasts cultured on double acid-etched titanium. Med Oral Patol Oral Cir Bucal. 2017 Sep 1;22(5):e651-e659. doi: 10.4317/medoral.22071. PMID: 28809380; PMCID: PMC5694190.

Di Carlo R, Di Crescenzo A, Pilato S, Ventrella A, Piattelli A, Recinella L, Chiavaroli A, Giordani S, Baldrighi M, Camisasca A, Zavan B, Falconi M, Cataldi A, Fontana A, Zara S. Osteoblastic Differentiation on Graphene Oxide-Functionalized Titanium Surfaces: An In Vitro Study. Nanomaterials (Basel). 2020 Apr 1;10(4):654. doi: 10.3390/nano10040654. PMID: 32244572; PMCID: PMC7221922.

Lumetti S, Manfredi E, Ferraris S, Spriano S, Passeri G, Ghiacci G, Macaluso G, Galli C. The response of osteoblastic MC3T3-E1 cells to micro- and nano-textured, hydrophilic and bioactive titanium surfaces. J Mater Sci Mater Med. 2016 Apr;27(4):68. doi: 10.1007/s10856-016-5678-5. Epub 2016 Feb 17. PMID: 26886816.

Petrini M, Pierfelice TV, D'Amico E, Di Pietro N, Pandolfi A, D'Arcangelo C, De Angelis F, Mandatori D, Schiavone V, Piattelli A, Iezzi G. Influence of Nano, Micro, and Macro Topography of Dental Implant Surfaces on Human Gingival Fibroblasts. Int J Mol Sci. 2021 Sep 13;22(18):9871. doi: 10.3390/ijms22189871. PMID: 34576038; PMCID: PMC8464951.

It is not clear if the disks have the same surface properties as commercially available implants.

AUTHOR’S ANSWER: Thank you for your observation. The discs have the same surface properties of the implants commercially available. The use of discs respect to the use of fixture in vitro studies permits researchers to evaluate the effects of superficial micro and nano topography without being influenced by the macro-geometry.

All the implant disks were supplied by a single manufacturer “AoN” which may indicate some bias towards that supplier.

AUTHOR’S ANSWER: Dear reviewer, thank you very much for your comment. The use of a single manufacturer permitted us to avoid the risk of bias connected with the influence of post-processing methods, that could influence the interaction between the cell and surfaces: the modality of sterilization, the packaging, and also the alloys used for the production of the discs.

The methods do not explain how the implant disks were sterilized, but later gives gamma radiation. Did the sterilization heating or gas or abrasion, modify the surface chemistry? Was an aseptic technique used when adding the implants to the cells?

AUTHOR’S ANSWER: Thank you for your observation. The discs were sterilized by Manufacturer exactly as the fixtures that are used for clinical activity; we have not reported the methods because they are classified as reserved information of the manufacturer. The discs arrived in our laboratory in self-sealing sterilization pouches and they have been opened in aseptic condition under a laminar flow hood using sterilized equipment.

The osteoblast cell lines were grown in DMEM culture with non-heat inactivated 10% fetal calf serum (FCS). Due to the use of FCS containing growth factors the native properties of the cells will be altered and they likely do not have the same bioreactivity of osteoblasts. Thus, it is likely that the cell responses are difference to natural osteoblasts, and this confounds the results.

AUTHOR’S ANSWER: Thank you for your observation. We used 10% of Fetal Bovine Serum (FBS), not calf. Furthermore, before the use the serum is subjected to heat inactivation at 56°C for 1 h, but we did not report this information in the manuscript.

The images are out of focus, pixilated, and indicate the wrong magnifications, and have text too small to read.

AUTHOR’S ANSWER: Thank you for your observation. We have modified the resolution of the images and the text in the figures.

There are too many problems with the quality of this article to recommend its publication. I suggest the authors be allowed to answer the reviewer comments and submit a revised article.

TITLE

  1. Acceptable

ABSTRACT

  1. Page 1, line 30. The incorrect statement must be deleted because no testing conformed to ISO standards for cytotoxicity: “None of the tested surfaces showed any cytotoxicity.”

AUTHOR’S ANSWER: Thank you for your suggestion. We have corrected the sentence and the abstract, following your suggestions.

INTRODUCTION

  1. The introduction needs to provide a background to the tests and results, so that readers can understand the topic. It failed, the introduction must explain the functions of osteoblasts, proliferation, biocompatibility, cytotoxicity, cell survival and bioactivity.

AUTHOR’S ANSWER: Thank you for your observations. We have introduced the following sentences:

“Osteoblasts are the key cells of the osseointegration process, promoting the deposition of bone matrix [Komori T, 2019]” (lines 67-68).

Furthermore, we have removed all sentences referred to cytotoxicity and biocompatibility assays. In this study we evaluated the proliferation, the adhesion, ALP activity and mineralization.

  1. The introduction needs to explain the purpose of this study, similar to the abstract. The purpose described here must be the same as the abstract, it is not and it confuses the reader.

AUTHOR’S ANSWER: Thank you for your suggestion. We have modified the aim in text of manuscript and we have corrected the sentence in “Indeed, the purpose of this study was aimed to compare the response of human oral osteoblasts on three different titanium surfaces: (i) Machined; (ii) double etched (Osteopore); (iii) double etched enriched with calcium and phosphorus (CaP) treatment (Nanopore) through the evaluation of cell proliferation, adhesion, interaction with Ti-discs, ALP activity and calcium deposition.”

MATERIALS AND METHODS

  1. There is a lack of controls to help validate the data.

AUTHOR’S ANSWER: Thank you for your observation. The control was represented by Machined discs. Machined disc is a surface in titanium GR4 with no roughening or CaP treatments applied

  1. It is not clear if the disks have the same surface properties as commercially available implants.

AUTHOR’S ANSWER: Thank you for your observation. The discs have the same surface properties of the implants commercially available. The use of discs respect to the use of fixture in vitro studies permit researchers to evaluate the effects of superficial micro and nano topography without being influenced by the macro-geometry.

  1. All the implant disks were supplied by a single manufacturer “AoN” which may indicate some bias towards that supplier.

AUTHOR’S ANSWER: Thank you very much for your comment. The use of a single manufacturer permitted us to avoid the risk of bias connected with the influence of post-processing methods, that could influence the interaction between the cell and surfaces: the modality of sterilization, the packaging, and also the alloys used for the production of the discs.

  1. The methods do not explain how the implant disks were sterilized, but later writes gamma ray exposure. Did the sterilization heating or gas or abrasion, modify the surface chemistry?

AUTHOR’S ANSWER: Thank you for your observation. The discs were sterilized by Manufacturer exactly as the fixtures that are used for clinical activity; we have not reported the methods because they are classified as reserved information of the manufacturer. The discs arrived in our laboratory in self-sealing sterilization pouches and they have been opened in aseptic condition under a laminar flow hood using sterilized equipment.

  1. For the scanning electron microscopy (SEM), why was x1,200 selected? It seems a low magnification for a SEM.

AUTHOR’S ANSWER: Thank you for your observation. The SEM analysis was performed with two magnifications: 290x (low magnification) and 1200x (high magnification) as reported in figure 1. In the methods there was a typing error, that we have corrected.

  1. The osteoblast cell lines were grown in DMEM culture with non-heat inactivated 10% fetal calf serum (FCS). Due to the use of FCS containing growth factors the native properties of the cells will be altered and they likely do not have the same bioreactivity of osteoblasts. Thus, it is likely that the cell responses are difference to natural osteoblasts, and this confounds the results. The use of FCS with growth factors must be justified and explained to readers.

AUTHOR’S ANSWER: Thank you for your observation. We used 10% of Fetal Bovine Serum (FBS), not calf. Furthermore, before the use the serum is subjected to heat inactivation at 56°C for 1 h, but we did not report this information in the manuscript.

  1. The proliferation study is not the same as biocompatibility/cytotoxicity, and there were no controls to help validate the proliferation data.

AUTHOR’S ANSWER: Thank you for your observation. We agree with your opinion, we removed all concerning the cytotoxicity and the biocompatibility in the manuscript. In this study was evaluated cell proliferation, adhesion, ALP activity and mineralization. Regarding the controls, in the proliferation assay we have reported two controls: seeded cells (104 cells) and cells cultured on the Machined disc.

  1. How does “multiphoton microscopy” differ from scanning electron microscopy (SEM)? This should be described as SEM to avoid confusing readers.

AUTHOR’S ANSWER: In this study the SEM has been used to evaluate the characterization of nano-topography of the discs without cultured cells. On the contrary, the multiphoton microscopy has been used to test cell adhesion. The adhesion was evaluated using DAPI, a fluorescent organic dye that highlights the nuclei of cells.

  1. Why were no controls used to validate the cell staining study? I disagree with the removal of the disks, because it likely discarded the attached cells.

AUTHOR’S ANSWER: Thank you for your observations. The control is represented by Machined disc. We removed the discs to allow taking the photo by microscope making visible the network created by the cells around the surface. In the presence of the discs this is no possible because light does not pass the surface.

  1. It is not clear if the cell culture media was changed during the 14 days of the ALP assay, and no control was used to help validate the accuracy of the data.

AUTHOR’S ANSWER: Thank you for your observation. The culture media has been changed every 3 days. We have introduced a sentence in the text “Briefly, 5x104cells/disc were seeded on Ti surfaces in 24-well culture plates for 14 days. Every 3 days fresh medium was added”

  1. I disagree with the removal of the disks from the Alizarin red staining, because it likely discarded the attached cells, and the lack of negative and positive controls to help validate the data.

AUTHOR’S ANSWER: Thank you for your observation. We removed the discs to observe the mineralization because AR staining was not visible on the top of the discs. The attached cells were not discarded, they were used for ALP assay. Furthermore, the positive control was represented by machined disc that is a surface without physical and chemical modifications. Machined surfaces were used as dental implants since 1970. While the negative control was represented by cells seeded on the plate (data not shown).

  1. The statistics were performed with ANOVA, but ANOVA only gives one P value per group. But the graphs show paired group P values, was this T tests? If so, the description of the statistics must be revised.

AUTHOR’S ANSWER: Thank you for you observation. We do not use the T-test, but ee used One Way ANOVA test and post hoc Tuckey test. We introduce the followed sentence in the methods:

“The Levene Test permitted the confirmation of the homogeneity of the groups tested; then the ANOVA and Tukey test was adopted”.

RESULTS

  1. The results state a 290x SEM magnification which is different to the x1200 SEM magnification in the methods. Which is correct or incorrect?

AUTHOR’S ANSWER: Thank you for your observation. The SEM analysis was performed with two magnifications: 290x (low magnification) and 1200x (high magnification) as reported in figure 1. In the methods there was a typing error, that we have corrected.

  1. How did the ANOVA P value give a P value between the three groups Nanopure, Osteopure, and Machined (P<0.001)

AUTHOR’S ANSWER: Thank you for your observation. We have added in the text the result of the One Way ANOVA test, that showed a p<0.0001. The single values of p-values among the groups were the results of the post hoc Tukey Test. We have added the followed sentences in the results:

“The ANOVA test resulted p<0.0001. Nanopore and Osteopore surfaces showed signifi-cantly larger contact angle than Machined (Machined vs Nanopore p=0.0001; Ma-chined vs Osteopore p<0.0001)  p<0.001). Thus, the wettability was altered by double etching and CaP treatments.”

“At the 5th day, the CaP-treated Nanopore surface showed a significant increase in cell population with +36.53 ± 0.13% relative to non-treated smooth Machined surface and showed greater number of cells +20.79 ± 0.13% than non-treated microrough Osteopore surface (Machined vs Nanopore p= 0.0001; Nanopore vs Osteopore p=0.001). The ANOVA test revealed a p=0.0008”

“Similarly, ALP was significantly stimulated when cells were seeded on Osteopore sur-face, despite a lower level was observed with respect to Nanopore (Machined vs Na-nopore p<0.0001; Machined vs Osteopore p<0.0001; Nanopore vs Osteopore p<0.0001) (Fig. 6). The ANOVA test resulted a p<0.0001”

DISCUSSION

  1. Unfortunately the discussion is as unfocused, confusing and as unhelpful as the introduction in helping the reader understand the topic and what was done and the significance of the results.
  2. Many of the sentences belong in the introduction as background.

AUTHOR’S ANSWER: Thank you for your suggestion. We have modified the entire text of discussion.

  1. p10, line 338: “Certain modifications” and “other modifications” have unclear meanings.
  2. p10, line 339: The final sentence is confusing and ambiguous to readers “This suggests that further time frames are important to analyze to fully understand the benefits of each variation.”

AUTHOR’S ANSWER: Thank you for your observation. We have modified the text of the manuscript:

“All together the results of this study suggest that physical and chemical modifications could show significant results at very short time after implant placement, however modifications might require between two and five days to obtain the osteoblast response. This suggests that the time frames are important to analyze and to fully understand the benefits of surface modifications. The results also suggest that Ti implants with combined nanoscale and microscale surface features have more favorable bone response than Machined. Moreover, the synergistic work between the nanotopography and the underlying microroughness results in a range of beneficial effects, and it is essential to consider at what time point and in what conditions these effects occur”.

CONCLUSION

  1. The conclusion is confusing; it should have explained the results more clearly to the reader. This can’t be the first step towards a novel surface, because the surfaces are already novel.

AUTHOR’S ANSWER: Thank you for your observation. We have modified the conclusion:

“The Nanopore surfaces, characterized by double etching treatment followed by CaP enrichment of grade 4 titanium surfaces, resulted in increased proliferation, adhe-sion, alkaline phosphatase activity and calcium deposition, compared to Machined and Osteopore disc. This study may represent an helpful step for clinicians in the vivo use of these the first step toward the individuation of a novel surface characterized by not only osteoconductive and, but also by osteoinductive properties”.

REFERENCES

  1. There are 19 references, most are recent, more references and more text in the introduction would improve the quality of this manuscript.

AUTHOR’S ANSWER: We have introduced other references:

  • Le Guéhennec, A. Soueidan, P. Layrolle, Y. Amouriq, Surface treatments of titanium dental implants for rapid osseointegration, Dent. Mater. 23 (2007) 844–854, https://doi.org/10.1016/j.dental.2006.06.025.
  • Mendonça, D.B.S. Mendonça, F.J.L. Aragão, L.F. Cooper Advancing dental implant surface technology–from micron- to nanotopography Biomaterials, 29 (2008), pp. 3822-3835, 10.1016/j.biomaterials.2008.05.012
  • Bandyopadhyay A, Mitra I, Shivaram A, Dasgupta N, Bose S. Direct comparison of additively manufactured porous titanium and tantalum implants towards in vivo osseointegration. Addit Manuf. 2019 Aug;28:259-266. doi: 10.1016/j.addma.2019.04.025. Epub 2019 May 1. PMID: 31406683; PMCID: PMC6690615.
  • Komori T. Regulation of Proliferation, Differentiation and Functions of Osteoblasts by Runx2. Int J Mol Sci. 2019 Apr 4;20(7):1694. doi: 10.3390/ijms20071694. PMID: 30987410; PMCID: PMC6480215.

IMAGES

  1.  The images are out of focus, pixilated, and indicate the wrong magnifications, and have text too small to read, as follows:
  2. Figure 1A, 1B, 1C and 1F are out of focus, and are not acceptable sharpness for publication
  3. Figure 1A. Nanopure is out of focus and not an acceptable sharpness for publication.
  4. Figure 2B. Bar chart is pixilated, and the font size is too small in the upper region.

AUTHOR’S ANSWER: Thank you for your observations. We have improved the resolution of the figures.

  1. Figure 2. Has no error bars, why?

AUTHOR’S ANSWER: The error bars are in the graphs, but the SD values were very small compared to the scale bar, thus the error bars were not visible. We reported in the caption the values of SD.

  1. Figure 4. I cannot see anything in most of these fluorescent images, and the blue cells are out of focus.

AUTHOR’S ANSWER: Probably there was a problem with the resolution of figures. We have improved them.

  1. Figure 5. The cell culture photographs are out of focus, and not sharp enough for a publication. The x6 magnification is incorrect, it must be about x200, you need to calculate the microscope magnification, the x6 is probably only the lens magnification.

AUTHOR’S ANSWER: These pictures were taken by a stereomicroscope connected with a camera. The lens magnification was 1x and the magnification was 6x. For the other images at higher magnification, 25x, the lens was 1x and the magnification was 25x. We have modified the methods: “Cells were then observed by a stereomicroscopy connected with a camera at a magnification of 6x and 25x (Leica, Wild Heerbrugg, Wetzlar, Germany)”.

  1. Figure 6. Has no error bars, why?

AUTHOR’S ANSWER: The error bars are in the graphs, but the SD values were very small compared to the scale bar, thus the error bars were not visible. We reported in the caption the values of SD.

  1. Figure 7. The cell culture photographs are out of focus, and not sharp enough for a publication. The x12 magnification is incorrect, it must be about x200, you need to calculate the microscope magnification, the x12 is probably only the lens magnification.

AUTHOR’S ANSWER: These pictures were taken by a stereomicroscope connected with a camera. The lens magnification was 1x and the magnification was 12x. Furthermore, there was a problem with the resolution of figures, we have improved them. We have modified the methods: “Images were taken by a  stereomicroscopy connected with a camera at a magnification of 12x (Leica, Wild Heerbrugg, Wetzlar, Germany)”.